# Big Data Analysis of Media Reports Related to COVID-19

**DOI:** 10.3390/ijerph17165688

**Published:** 2020-08-06

**Authors:** Ji-Hee Jung, Jae-Ik Shin

**Affiliations:** 1Department of Business Administration, University of Ulsan, 93 Daehak-ro, Nam-gu, Ulsan 44610, Korea; aboutjee@ulsan.ac.kr; 2Department of Distribution, Gyeongnam National University of Science and Technology, 33 Dongjin-Ro, Jinju, Gyeongnam 52725, Korea

**Keywords:** COVID-19, big data, government countermeasure, Korea

## Abstract

COVID-19 is lasting longer than expected, which has a huge impact on the economy and on personal life. Each country has a different response method, and the damage scale is also distinct. This study aims to find out how COVID-19-related news was handled in the domestic media to seek ways to minimize the pandemic. The paper focuses on the number of news features by period and by disaster and analyzes related words based on big data. The results of the analysis are as follows. First, in the initial response phase, keywords to identify accurate sources of actual broadcast contents, fake news, social networking service (SNS), etc. were also ranked in the top 20. Second, in the active response phase, when the number of confirmed persons and the government’s countermeasures were announced, more than 100 COVID-19-related articles were issued, and the related words increased rapidly from the initial response stage. Therefore, the fact that COVID-19 has been expressed as a keyword indicates that our society is watching with great interest in the government’s response to the disease.

## 1. Introduction

Since the 35-year-old woman of Chinese nationality who arrived in Wuhan city, China, on January 20 was found as the first confirmed case, the number of newly infected coronavirus infections in Korea (hereinafter referred to as COVID-19) is growing scary. The increase in the number of confirmed cases that had once entered a calm phase was triggered by the occurrence of the 31st confirmed patient on 18 February 2020. On February 26, about a week later, the number of confirmed patients surpassed 1000, and two days later, on February 28, the number doubled.

The World Health Organization (WHO), which continued its passive action as the worldwide spread of COVID-19 continued, starting with China and neighboring Asian countries, finally declared a “pandemic (global pandemic)” for COVID-19. “Pandemic” is a term that describes the sixth stage of the highest risk of the WHO’s pandemic alert rating and refers to the global spread of an epidemic.

COVID-19 has infected millions of people around the world. Countries try to take effective measures to combat the pandemic; prevent the spread; decrease the risk of infection; restore the economy, protect the society, and preserve the environments; and make up for the losses caused during the global epidemic [1,2]. In this kind of the pandemic situation, many nations have the capacity to cope with tangible and intangible risks resulting from the virus and take appropriate the prevention measures [2]. Since the medical system differs from country to country and government policies are not the same, it can be a meaningful study by tracking the changes in key keywords of the media for the Korean government’s response policy. This study period is limited from January 20, 2020 to April 30, 2020.

Meanwhile, the rapid spread of COVID-19 and worldwide interest is leading to a surge in media reports related to COVID-19. As the public mainly acquires limited information on the media and recognizes the information by adding personal feelings and experiences in it, in order to understand the public’s perception of COVID-19, it is necessary to look at domestic media reports related to it. Many citizens also have desires to get a lot of information from the various media to figure out the situation, and to protect their health. Information-seeking behaviors can decrease their anxiety triggered by uncertainty during the disaster [3,4]. However, this paper focuses on keyword changes in newspapers, magazines, and broadcasts, excluding the Internet.

This study aims to find out how COVID-19-related news was handled in the domestic media. We used BIGKinds, which provides big data analysis technology and integrated data composed of news collected from various media companies such as newspapers and broadcasters. In particular, this paper focuses on the number of news features by period and by disaster and analysis of related words based on big data. Through this, we want to track the public interest in COVID-19 and the government’s policy changes and seek ways to minimize the pandemic.

## 2. Theoretical Background 

### 2.1. COVID-19

The official name of the disease is Coronavirus Infection-19 (COVID-19), which refers to a viral respiratory disease that occurred in Wuhan, Hubei Province, China, in December 2019. SARS (Severe Acute Respiratory Syndrome), which was first known as a respiratory epidemic of unknown cause, but was later determined to be spread by the pathogen in 2003, has been identified as a coronavirus, similar to the prevalence of MERS (Middle East Respiratory Syndrome) in 2012. The disease often attacks the respiratory system and develops into pneumonia after symptoms such as high fever, sore throat, cough, and difficulty breathing. The incubation period is about 3 to 7 days, but can last up to 14 days or more, and is known to have the highest initial propagation power with mild symptoms. COVID-19 is a virus created by mutation of an existing virus while parasitic on another host. It cannot be said to be a new virus, but it causes fear because there is no treatment or vaccine.

COVID-19 is sparking off an unknown serious international public health emergency [5] since it has shown a wide range of symptoms, from mild flu-like prodrome such as cold, sore throat, cough and fever, to more severe features such as pneumonia and breathing difficulties, and in some cases, death [6]. This shows that the world will have many difficulties in coping with COVID-19, and the vaccine development is not expected to be easy. In addition, 

The occurrence of COVID-19 can be categorized as a typical global crisis pandemic, which is defined as a specific and surprising event, leading to high levels of uncertainty and serious threats [7]. According to Worldometer, a real-time international statistics website, as of 25 April 2020, at 10:56 a.m., the number of COVID-19 cumulative diagnoses worldwide was 2,830,051 (+111,352). The number of deaths was 197,245 (+6,591), and the recovered number was 798,772. As of 25 April 2020, Korea (31st) had 10,718 confirmed patients (+10) and 240 deaths (+0).

Table 1 shows the spread process of COVID-19 until April 30, after the first confirmer in Korea came out on January 20, 2020. Figure 1 also shows the cumulative status of new confirmers and accumulated ones in Korea from January 1 to April 30, 2020. On February 29, 2020, the number of the confirmers reached 909 and from April 6, 2020, it fell to 50 or fewer. Since April 18, 2020, it fell to less than 20. This change in the number of confirmed persons will identify the Korean government’s COVID-19 response policy. Thus, this study examines how the trend of media coverage has changed according to the spread process of COVID-19 and the related issues and may provide the implications.

### 2.2. Utilization of Big Data in the Media

According to the theory of media dependence [8], during a serious social turmoil, the demand for related information and situational awareness is unusually high, and the media are generally recognized to best meet these needs [9]. Specifically, the public depended heavily on the media to get information on the reaction of organizations [10,11], and also on an exchange of opinions with others [12].

In the last two decades, new media, such as social media and YouTube based on advanced Internet communication technology, have emerged, gradually replacing traditional media with their advantages such as convenience, time and cost [13]. People have mainly employed social media to communicate with the public and obtain crowd-sourced information [14]. Therefore, the new media can have an effective impact on informing people of the severity of COVID-19. 

In modern society, data are rapidly increasing due to the development of numerous information technologies and the use of social media [15], and the concept of big data began to appear to create new meaning and value from the vast amount of existing data [16]. Big data can be defined as information technology to predict the future by generating valuable information based on large amounts of data practically and efficiently. It has a characteristic that is different from existing data in terms of size, speed, and type of data [17]. It also means a large set of vast data sets exceeding the range that a typical database can store, manage, and analyze. 

Newspaper data have been extensively studied in the field of media and information science. However, research methods that mainly read one-to-one and analyze the contents have been carried out rather than mechanical text mining. Recently, with the development of data mining tools, many papers analyzing newspaper articles and public opinion data have begun to be published [18]. Press big data refer to the result of extracting unstructured data into structured meta-data such as institutions, numbers, people, and quotations through image processing, natural language processing, and semantic network analysis. It has not only the unstructured nature of the text form itself but also the structured characteristics of media, genre, and date. Therefore, it is relatively easy to handle and manage than the completely unstructured data [19]. Therefore, many studies have been conducted using press big data in various academic fields. Through the big data analysis, we can examine various trends of the present and past in society. The data will be materials that can be researched in depth while tracking an issue and will have the function of finding a hidden context of information in the relationship network analysis [20].

The BIGKinds service is a news reporting analysis service evolved from a news search service by KINDS (Korea Integrated News Database System). The KINDS service is a service that tracks various news such as broadcasts and major daily newspapers, which began in 1990 and provides search services. Existing KINDS service utilizes the big data analysis technology, which has recently been spotlighted on the vast amount of news data accumulated so far, while the novel news information provision and search system” BIGKinds” was established on April 19, 2016 and the service has grown and come so far [21]. The BIGKinds service consists of news collection, analysis, and storage systems, and as of May 2020, everyone can search and use 10 million cases published by fifty-four media outlets (eleven national newspapers, eight economic newspapers, twenty-eight local newspapers, five broadcast stations, and two specialty magazine).

## 3. Research Method 

### 3.1. Analysis Target

In this study, through the “BIGKinds”, a news big data analysis system from the Korea Press Foundation, media reports were classified and analyzed according to keywords. From 20 January 2020 to 30 April 2020, the search keyword covering new coronavirus infection, coronavirus infection, new coronavirus, coronavirus, and COVID-19 was “corona”, the purpose of this study was to examine the changes and implications of issues related to the keywords.

The scope of the study was based on the current trend of COVID-19 and the government’s response process. The period of response after COVID-19 was introduced into Korea (20 January 2020 to 17 February 2020) is called the “initial response stage”. After the appearance of the 31st patient, the first case of infection in the Shincheonji Church in Daegu (from 18 February 2020), the period of response, which the government raised to “the highest level of infectious disease crisis warning”, is classified as the “active response stage”. 

We analyzed twenty-two media companies including three broadcasting stations (KBS, MBC, and SBS) and news specialized channel (YTN), ten national newspapers (Kyunghyang Newspaper, Kookmin Daily, Dong-A Daily, Culture Daily, Seoul Newspaper, World Daily, Chosun Daily, JoongAng Daily, Hankyoreh, and Hankook Daily), and eight economic newspapers (Maeil Economy, Money Today, Seoul Economy, Asian Economy, Ajou Economy, Financial News, Korea Economy, and Herald Economy). The articles were classified into political, economic, social, cultural, regional, and IT science. Incident classification also included all media reports related to crime, accidents, disasters, and society. 

### 3.2. Data Collection Standard and Procedure

The importance of big data is increasingly emphasized in various research fields, and it is used as a method to efficiently solve complex and diverse problems [22]. The BIGKinds program provides a function to calculate the frequency of words appearing in related text and convert them into a visualized image according to their weight and frequency. In this study, keyword trends and related words were analyzed. The trends were researched by counting the number of article occurrences and displaying graphs daily in the news searched by keywords. The data searched by the keyword are analyzed in the association with the topic rank algorithm and displayed in the word cloud form [23].

The word cloud is a technique that extracts keywords from a document and visualizes them so that the nature of the document can be checked intuitively. After the extraction of keywords in the news set, by a researcher, and analysis of the correlation between them using the topic rank algorithm, they are visualized and displayed by their degree of relevance in a word cloud. The topic rank is an analysis method that extracts related keywords by calculating the occurrence frequency and importance of them expressing simultaneously with a specific keyword and is commonly used when educing key concepts [24]. Therefore, in this study, the media coverage of “corona” was visualized and the related topics were investigated to analyze the media report type, focusing on issues derived from the weight.

## 4. Research Results 

The subject of this study was to classify and analyze press reports according to keywords. From 20 January 2020 to 30 April 2020, the search keyword had a total of 7719 news features about “COVID-19”, and 7391 cases were finally used for the analysis through filtering. The results are shown in Figure 2.

### 4.1. Initial Response Phase (20 January 2020–17 February 2020)

In the case of the first confirmed patient in Korea on January 20, the Chinese woman entered Korea, the government raised the level of the infectious disease crisis warning from “interest” to “attention”. Then, the government increased the level of warning of infectious diseases from “attention” to “border” on January 27 and started operating the Korea Centers for Disease Control and Prevention (KCDC) to handle new coronavirus infections. On February 3 (143 cases) and February 4 (142 cases), articles related to “corona” showed an increasing trend, and the government restricted the entry of foreigners from Hubei Province, China, and carried out the measurements of the “special entry procedure” for immigrants and temporary suspension of entry to Jeju-do visa.

In all, 1638 articles (excluding analysis: 96 cases) were extracted during the initial response phase (20 January 2020 to 17 February 2020), which is the time to respond after COVID-19 has spread to Korea. The results are shown in Figure 3. And Table 2 shows the related keywords ranking and the word cloud in the initial response phase (January 20, 2020-February 17, 2020).

The number of COVID-19-related articles has exploded on the outbreak of domestic confirmed case. China, China Wuhan, Wuhan, and China Hubei, which are keywords for the place where COVID-19 occurs, were located at the top. As the uncertain information about COVID-19 spread along with the articles related to the government response (KCDC and President Moon Jae-in), keywords to identify the accurate sources of actual broadcast contents, fake news, and SNS, etc. were also ranked in the top 20. Fake news that tried to arouse interest by using the public’s anxiety was shared with the Internet, causing confusion.

Moreover, interest in association with SARS and MERS, respiratory infections that had been prevalent in the past, was also high. The number of COVID-19 diagnosed and deceased patients passed that of SARS (severe acute respiratory syndrome) in 2003, at that time, more than 5327 people were confirmed with SARS in mainland China, of which 349 were killed. As it reflected this, SARS was ranked at the top of the list of major new coronavirus keywords.

### 4.2. Active Response Phase (18 February 2020–30 April 2020)

After the first cases of infection in the Shincheonji Church, the number of confirmed cases increased rapidly, mainly in Daegu and Gyeongbuk, and the government raised the alert on the infectious disease crisis to the highest level of severity.

The frequency of articles due to the outbreak of clusters of Shincheonji in Korea was the highest with 130 cases on February 24, and the World Health Organization’s (WHO) pandemic declaration on March 11 saw 127 corona-related articles surge. Hundreds of confirmed cases occurred every day, and on March 3, the cumulative number of diagnoses in Korea reached 5000, and on April 3, 10,000. On March 22, high-intensity social distancing began, and on April 4, it was extended by two weeks. On April 19th, the intensity of social distancing was eased and extended until May 5th, and there was no daily death on April 24. Whenever the number of confirmed persons and the government’s countermeasures were announced, more than 100 corona-related articles were issued, and related words increased rapidly from the initial response stage.

In all, 5753 cases (excluding analysis: 232 cases) were extracted in the active response phase (18 February 2020–30 April 2020) after the emergence of the 31st patient, the first case of infection in the Shincheonji Church in Daegu, which became the watershed in Korea. The results are shown in Figure 4. And Table 3 shows the related keywords ranking and the word cloud in the aggressive response phase (February 18, 2020–April 30, 2020).

In the aggressive response phase, the number of keywords (Confirmed case and People) related to this was rising as the number of domestic confirmed cases surged. Furthermore, instead of keywords related to the origin of China, Daegu and Shincheonji, which influenced the spread in Korea, were ranked third and sixth, respectively. On March 11, the World Health Organization (WHO) declared a pandemic when signs of spread to Europe and signs of a pandemic were seen. Related keywords (Europe and Pandemic) were ranked in the top. As the secondary infection in the community began, keywords (Our Country, Korea, People and Local Government) also appeared mainly nationwide and in regions.

President Moon declared South Korea to be “the number one country that had coped well with the pandemic.” Thanks to domestic and foreign media praise of the “South Korean model,” the ruling Korea Democratic Party (and its satellite party) captured an unprecedented, near two-thirds majority of the National Assembly (180 of 300 seats) on 15 April 2020, and Moon’s approval rating rose to as high as 70% [25]. This shows that Koreans have well accepted and followed the government’s COVID-19 response policy.

Since March 22, social distancing has been reinforced, and related keywords have increased. From March to the beginning of April, the outbreak spreading trend continued, but from April 5, when the social distancing was in place for over a month, the spreading rate slowed significantly. Companies participated in flexible work and video conferences, and citizens actively participated in social distancing by canceling meetings, taking food deliveries rather than eating out, and changing consumption behavior through online shopping. Thus, it is identified that reducing personal contacts is a central measure against the spreading of the novel coronavirus disease.

## 5. Conclusions

In this study, keyword trends were analyzed based on media reports related to COVID-19 using big data analysis. The implications obtained based on the research results are as follows. First, it was confirmed that the interest in the people or society through keywords was changed according to the situation of COVID-19. In the early stages of the outbreak and epidemic, it was confirmed that the interest in origins such as China and Wuhan was high, but as the spread of COVID-19 became full-scale, the interest in Shincheonji, Daegu, and domestic clusters increased.

Second, through the analysis of big data, it was found that COVID-19 affects the daily lives of individuals and elicits a high interest in the government response process. The degree of interest in the government response process was confirmed by the fact that “President Moon Jae-in” has emerged as a key keyword in the entire response phase. The fact that COVID-19 has consistently occupied a high frequency since its inception in Korea and has been expressed as a keyword indicates that our society is watching with great interest in the government’s response to the disease.

Third, from April 5 to April 30, it was identified that the government’s high-intensity policy of social distancing suppressed the increase in the number of COVID-19 confirmed case in Korea. As the United States or Europe loosened the social distancing, the number of confirmed cases exploded. Thus, high-intensity social distancing can be one way to help minimize COVID-19 events.

This study is a basic study that checks the degree of COVID-19′s domestic news publication and related words. It is significant that the results of this study confirmed the interest in COVID-19 and the weight of related keywords even at the basic level. However, it is at the point when COVID-19 has not completely ended, and it has the limitation that it is a result of a partial search using the keyword “corona”. In future studies, it is necessary to expand the trend of related keywords and terms related to COVID-19 and to develop the research method, such as identifying issues focused on the main word. It is also required to develop a big data analysis that confirms whether citizen awareness and government policies in response to COVID-19 are effective. 

## Figures and Tables

**Figure 1 ijerph-17-05688-f001:**
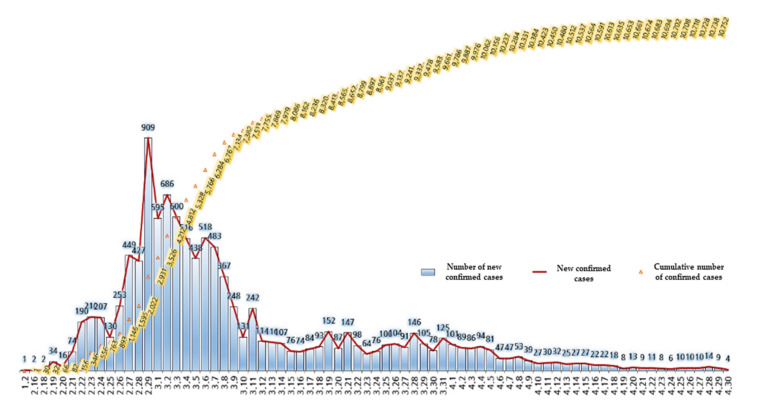
Daily status of new and cumulative confirmed cases in Korea.

**Figure 2 ijerph-17-05688-f002:**
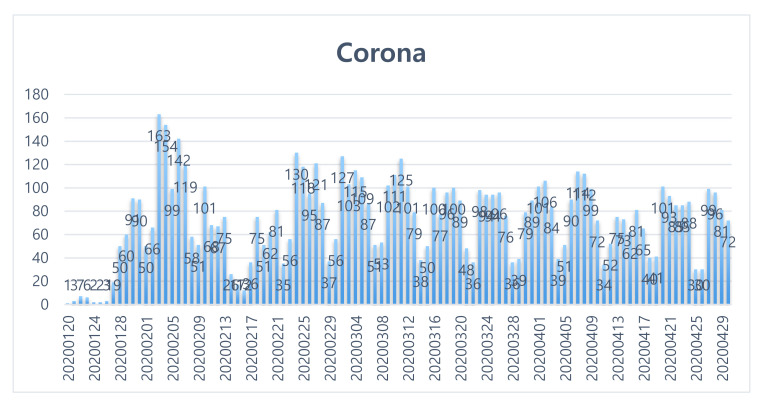
Media coverage trends related to “corona” (all articles).

**Figure 3 ijerph-17-05688-f003:**
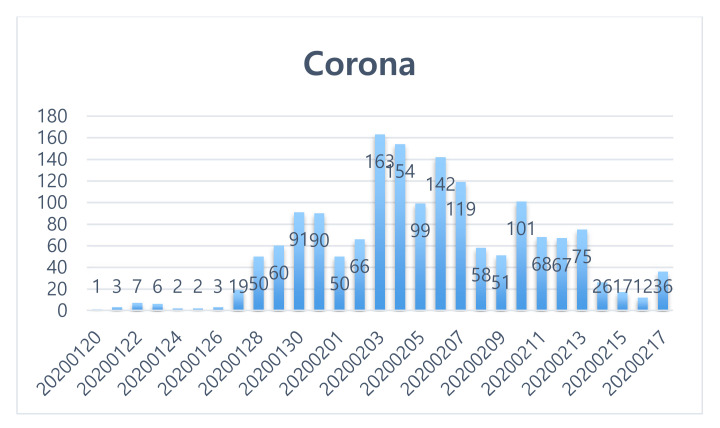
Keyword trends (initial response stage: 20 January 2020~17 February 2020).

**Figure 4 ijerph-17-05688-f004:**
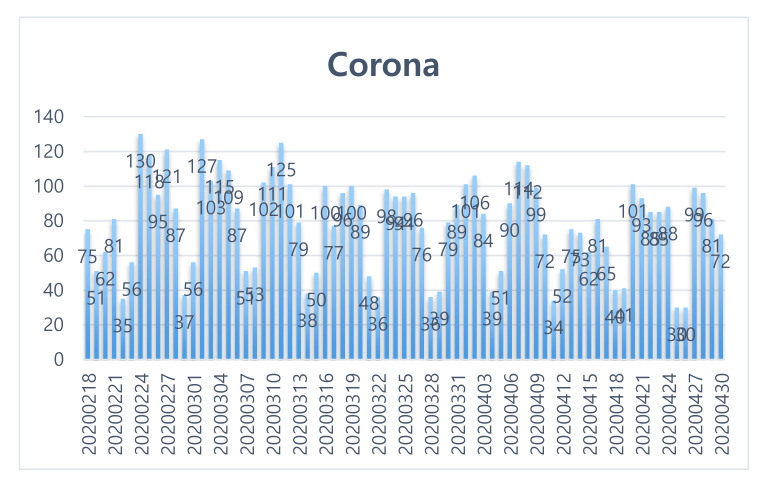
Keyword trends (active response stage: February 17 2020~30 April 2020).

**Table 1 ijerph-17-05688-t001:** COVID-19-related issues in Korea.

Date (D-M-Y)	Issues
20 January 2020	The first confirmed case of COVID-19 in Korea. A 36-year-old Chinese woman who arrived in Incheon on the 19th from Wuhan, China: Infectious disease crisis alert level up from “Attention” to “Caution”.
24 January 20	The Second confirmed case, a 56-year-old Korean man who arrived at Gimpo Airport from Wuhan.
27 January 2020	Infectious disease crisis alert level up from “Caution” to “Warning”: New coronavirus central accident control headquarters in operation.
28 January 2020	January 13~26 to begin a full survey of immigrants from Wuhan.
31 January 2020	The first returning Korean residents from Wuhan start of isolated life at temporary living facilities in Jincheon and Asan.
4 February 2020	Restrictions on the entry to foreigners from China’s Hubei province. Application of “special entry procedures” for immigrants from China. Suspended international visa-free entry to Jeju Island.
5 February 2020	Second patient discharged. The first discharge of a domestic patient in 13 days of confirmed diagnosis.
12 February 2020	New coronavirus infections are named “COVID-19” according to the World Health Organization (WHO)’s decision to name diseases.
18 February 2020	Confirmed diagnosis 31st patient (61-year-old female, Korean), the first case of infection in Shincheonji Church in Daegu.
19 February 2020	A full survey of more than 1000 believers who participated in the worship service of the Shincheonji Church in Daegu with 31st patient.
20 February 2020	The first death occurred. A confirmed case related to Cheongdo Daenam Hospital (63-year-old male).
23 February 2020	Raise the infectious disease crisis alerted to the highest level of “Serious”.
25 February 2020	With the cooperation of the Shincheonji Church, the entire COVID-19 investigation will begin.
26 February 2020	The cumulative number of confirmed cases in Korea is 1146, reaching 1000.
29 February 2020	909 new confirmed cases per day, the largest increase in scale.
2 March 2020	The first opening of a life therapy center for mild patients in Daegu.
3 March 2020	The cumulative number of confirmed cases in Korea is 5186, reaching the 5000 mark.
7 March 2020	Implementation of a five-part mask system that determines the date of purchase according to the year of birth.
8 March 2020	The first confirmed case of the Seoul Guro Call Center, the largest case of collective infection in the metropolitan area (56-year-old female).
11 March 2020	World Health Organization (WHO) declared a global pandemic on COVID-19.
15 March 2020	The declaration of special disaster zones in Daegu and parts of North Gyeongsang Province: “Special entry procedures” will be applied to those departing from France, Germany, Spain, Britain, and Netherlands and will be expanded to those entering Europe on the 16th.
22 March 2020	Start of “high-intensity social distancing” such as restrictions on the operation of religious, entertainment, and indoor sports facilities: Mandatory examination of immigrants from Europe, mandatory self-isolation for two weeks.
27 March 2020	Mandatory self-isolation for two weeks from the United States.
1 April 2020	Self-isolation required for all immigrants for two weeks.
3 April 2020	The cumulative number of confirmed cases in Korea is 10,062 people, reaching the 10,000 mark.
4 April 2020	Announcement of two weeks extension of high-intensity social distancing.
13 April 2020	Mandatory examination of immigrants from the United States.
15 April 2020	Voting of the National Assembly, including COVID-19, confirmed and self-isolated cases.
18 April 2020	18 new confirmed cases. Decreased confirmed cases.
19 April 2020	Announcement of extension of May 5th by mitigating social distancing.
24 April 2020	Zero daily deaths in 39 days.

**Table 2 ijerph-17-05688-t002:**
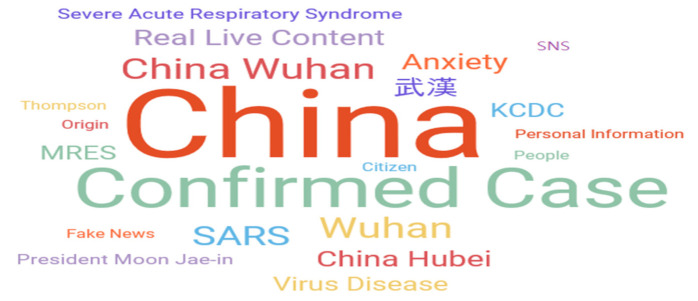
Related keywords (initial response stage: 20 January 2020~17 February 2020).

Rank	Keyword (Weight)	Rank	Keyword (Weight)
1	China (37.52)	11	Virus Disease (8.00)
2	Confirmed Case (22.02)	12	KCDC (7.50)
3	China Wuhan (12.00)	13	President Moon Jae-in (5.87)
4	Wuhan (11.73)	14	Severe Acute Respiratory Syndrome (5.83)
5	SARS (11.12)	15	Fake News (5.54)
6	Real Live Content (9.33)	16	Personal Information (5.54)
7	Wuhan (8.87)	17	People (5.33)
8	Anxiety (8.73)	18	Social Networking Service (SNS) (5.33)
9	China Hubei (8.43)	19	Origin (5.31)
10	MERS (8.19)	20	Citizen (4.67)

KCDC: Korean Centers for Disease Control and Prevention; SARS: Severe Acute Respiratory Syndrome; MERS: Middle East Respiratory Syndrome.

**Table 3 ijerph-17-05688-t003:**
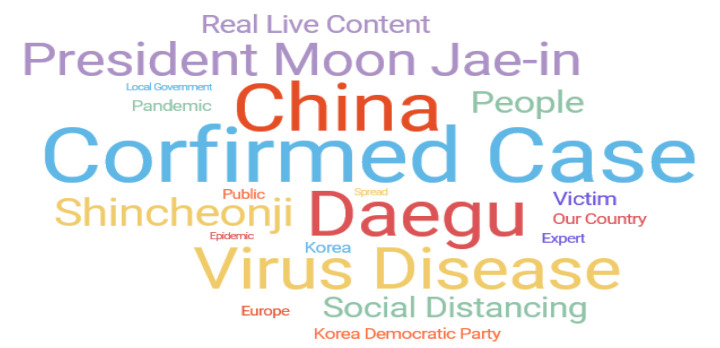
Related keywords (active response stage: 17 February 2020~30 April 2020).

Rank	Keyword (Weight)	Rank	Keyword (Weight)
1	Confirmed Case (20.60)	11	Korea Democratic Party (6.69)
2	China (18.57)	12	Pandemic (6.67)
3	Daegu (17.87)	13	Our Country (6.55)
4	Virus Disease (16.43)	14	Korea (6.55)
5	President Moon Jae-in (13.74)	15	Expert (6.21)
6	Shincheonji (11.92)	16	Public (6.12)
7	People (10.21)	17	Europe (6.00)
8	Social Distancing (9.92)	18	Epidemic (5.31)
9	Real Live Content (8.67)	19	Spread (5.19)
10	Victim (7.54)	20	Local Government (5.04)

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
