# Peer review of "Big Data Analysis of Media Reports Related to COVID-19"

_ijerph, 2020, doi:10.3390/ijerph17165688_

Round 1

Reviewer 1 Report

This paper presents a study of COVID19 in Korea based on the number of news by period, the number of news by disaster, and analyzes related words based on big data.

However, the research proposal and method to analyze is not satisfying and needs more clarification.

  1. How big data was used to analyze news reports?
  2. What model has been used to analyze this data?
  3. How proposed analysis provide more details about COVID spread?
  4. In figure 1 what is the yellow band of line?
  5. Most importantly, authors have not proposed the research problem in a coherent way. This research lacks lot details about the analysis procedure and conclusion that can be made from it.

Author Response

Thank you for your careful review. We revised and supplemented the points you pointed out as follows.

  • Big Kinds, used in big data analysis in this study, is a new news information delivery and search system that utilizes the big data analysis technology that has recently been in the spotlight for vast amounts of news data. In the past, text mining has been mainly conducted by research methods that read one-on-one and analyze contents. Recently, with the development of data mining tools, papers analyzing large volumes of newspaper articles and public opinion data have begun to be published, and this research has also been used to study the topic of Corona19. I also added an explanation and presented a reference to the research.
  • The media coverage analysis of this study confirmed that the public or society's interest through keywords changes depending on the trend of Corona19. In the early days of the outbreak and trend, interest in the origin of China and Wuhan was high, while the spread of Corona 19 was in full swing, it was confirmed that interest in the domestic regions and clusters such as Sincheonji and Daegu increased. Big data analysis shows that Corona 19 affects each individual's daily lives and has a high level of interest in the government's response process.
  • The yellow line in Figure 1 means the Cumulative Number of Confirmed Cases and is shown in the plot.

Reviewer 2 Report

Big Data Analysis of Media Reports Related to 2
Covid-19

Comments and Suggestions for Authors
The work seems to me very pertinent, timely and necessary.
For this reason alone, I consider your post to be positive.
However, I believe that major changes need to be made first in
at least two parts of this document.
1. All the figures used are not the most appropriate to visualize
the results. They are incomprehensible and illegible. Other
infographics should be used.
2. The bibliography seems insufficient and it is also
alphabetically disordered. I suggest exploring works on the
perspective of communication and public health, where there is
a trajectory and references of great interest.

Author Response

Thank you for your careful review. We revised and supplemented the points you pointed out as follows.

  • Big Kinds, used in big data analysis in this study, is a new news information delivery and search system that utilizes the big data analysis technology that has recently been in the spotlight for vast amounts of news data. In the past, text mining has been mainly conducted by research methods that read one-on-one and analyze contents. Recently, with the development of data mining tools, papers analyzing large volumes of newspaper articles and public opinion data have begun to be published, and this research has also been used to study the topic of Corona19. I also added an explanation and presented a reference to the research.
  • The word cloud form of Table.2 and Table.3 has been modified to match the figure of Table.
  • The reference has been supplemented and revised.

Reviewer 3 Report

The theoretical framework is adequate. It conforms to the proposed objectives. The same applies to the applied methodology, in accordance with the objectives and conclusions and results.

Author Response

Thank you for your careful review. The revised paper is supplemented and loaded.

Reviewer 4 Report

Dear authors,

This study aims to find out how news related to Covid-19 was handled in the domestic media to look for ways to minimize the pandemic. It is very important and interesting to analyze the reports on COVID that the newspaper reported. However, the study presents some problems that must be solved.

  • Authors need to improve the presentation of results in the figures 2, 3 e 4. It is not clear. The references of the figures are not in the text.
  • Authors need to present the statistical correlations between the news and the confirmed cases of Covid-19. Only the description of the facts in table 1 does not support the results found.
  • The study focuses on the number of news by disaster, it is not clear what it means.
  • What is the criterion used to associate the keywords in the word cloud (table 3)? They have the same weight? Examples: Confirmed Case, Korea Democratic Party, Virus Disease.
  • Lines 101-106: the authors analyzed twenty-two media companies including three broadcasting stations. The authors should show results found by each media company. This can make a difference in the analysis of results
  • Lines 106-108: The articles were classified into political, economic, social, cultural, regional, and IT science. Incident classification also included all media reports related to crime, accidents, disasters, and society. It is not clear how these data were presented and discussed in the study

If the authors can make the revisions to improve the pointed parts, the manuscript will be worth publishing in the journal

Author Response

Thank you for your careful review. We revised and supplemented the points you pointed out as follows.

  • The description of Figures 2, 3 and 4 has been added to the body.
  • The word cloud form of Table.2 and Table.3 has been modified to match the figure of Table.
  • Big Kinds, used in big data analysis in this study, is a new news information delivery and retrieval system that utilizes the latest big data analysis technology that has been in the spotlight for vast amounts of news data. In the past, text mining has been mainly conducted by research methods that read one-on-one and analyze contents. Recently, with the development of data mining tools, papers analyzing large volumes of newspaper articles and public opinion data have begun to be published, and this research has also been used to study the topic of Corona19. I also added an explanation and presented a reference to the research.
  • The reference has been supplemented and revised.

Round 2

Reviewer 1 Report

My concerns were addressed.

Author Response

Thank you for your careful review.
The review paper corrected grammar and typos through correction.
The revised content in the examination paper is marked in red.
We will supplement and mount the revised paper.
As you advised, we will strive for better research.